# Impact of Fe and Ni Addition on the VFAs’ Generation and Process Stability of Anaerobic Fermentation Containing Cd

**DOI:** 10.3390/ijerph16214066

**Published:** 2019-10-23

**Authors:** Huayong Zhang, Yanli Xu, Yonglan Tian, Lei Zheng, He Hao, Hai Huang

**Affiliations:** Research Center for Engineering Ecology and Nonlinear Science, North China Electric Power University, Beijing 102206, China; 17600754732@163.com (Y.X.); yonglantian@ncepu.edu.cn (Y.T.); 15600462123@163.com (L.Z.); haohe333@126.com (H.H.); huanghai@ncepu.edu.cn (H.H.)

**Keywords:** compound heavy metals, anaerobic fermentation, biogas properties, process stability, substrate biodegradation, microbial properties

## Abstract

The effects of Cd, Cd + Fe, and Cd + Ni on the thermophilic anaerobic fermentation of corn stover and cow manure were studied in pilot experiments by investigating the biogas properties, process stability, substrate biodegradation, and microbial properties. The results showed that the addition of Fe and Ni into the Cd-containing fermentation system induced higher cumulative biogas yields and NH_4_^+^–N concentrations compared with the only Cd-added group. Ni together with Cd improved and brought forward the peak daily biogas yields, and increased the CH_4_ contents to 80.76%. Taking the whole fermentation process into consideration, the promoting impact of the Cd + Ni group was mainly attributed to better process stability, a higher average NH_4_^+^–N concentration, and increased utilization of acetate. Adding Fe into the Cd-containing fermentation system increased the absolute abundance of *Methanobrevibacter* on the 13th day, and *Methanobrevibacter* and *Methanobacterium* were found to be positively correlated with the daily biogas yield. This research was expected to provide a basis for the reuse of biological wastes contaminated by heavy metals and a reference for further studies on the influence of compound heavy metals on anaerobic fermentation.

## 1. Introduction

Energy is the driving force for industrial development and global economic growth. In China, energy plays a key role in supporting one-fifth of the world’s population and maintaining fast-growing gross domestic product (GDP) [1]. Similar to the case in a lot of other countries, fossil fuels are the main source of fuel for various forms of energy utilization in China [2]. However, fossil fuel resources are limited and will produce emissions of carbon dioxide (CO_2_), nitrogen oxides (NO_x_), and other harmful gases in the process of utilization, which will cause environmental pollution and human health risks [1]. Consequentially, the world is looking for alternative, renewable, and sustainable energy sources [3]. Bioenergy is one of the most promising renewable clean energy options [4]. Biomass, including bioenergy plants, as a large, clean, and renewable energy, can effectively alleviate the energy shortage and reduce the emission of harmful gases [5].

Anaerobic fermentation, which uses biomass to produce biogas as clean energy, is an environmentally friendly and promising energy production technology [6]. The anaerobic fermentation process involves different types of microorganisms, and the growth and metabolism of those microorganisms are affected by various trace metal elements [7]. High levels of heavy metals are toxic and non-degradable, and hence are threats the environment safety [8]. A previous study demonstrated that trace metal elements (such as Cu, Ni, Zn, Cd, Cr, and Pb) had a certain inhibitory effect on anaerobic digestion at a high concentration [9]. Heavy metals had cytotoxic effects and were significant in biochemical reactions. They affected the activity of enzymes and microorganisms, and then impacted the biogas production efficiency of anaerobic fermentation [10]. A majority of studies showed that the toxic effect of trace heavy metals, Cr, Cd, and Ni, on anaerobic fermentation was because of the combination of metal ions with thiols and other groups on proteins, which destroyed the structure and activity of enzymes [9,11,12].

The demand for heavy metals may vary in different stages of anaerobic fermentation, and methanogenesis is the most metal-rich enzymatic pathway [13]. However, the demand trend for metals in all stages is basically the same: Fe is the most abundant metal, followed by Ni and Co, with a small amount of Mo and Zn [14]. Tian et al. found that Cd and Ni were beneficial to biogas production in anaerobic fermentation at low concentrations [15]. The addition of Fe effectively resulted in the variation of cellulase during anaerobic fermentation, and then increased the cumulative biogas production (up to 18.1%) and CH_4_ content (up to 8.3%) [16]. Ni played an important role in the process of anaerobic digestion, mainly because the growth of all methanogens and the synthesis of cofactor F_430_ needed the participation of Ni [17]. It was found that the concentration of dissolved Ni over 1 mg/L would inhibit the methanogenesis process through the toxic metabolic of Ni tolerant degradative organisms in the process of anaerobic fermentation of sludge [18]. Another study showed that the concentration of Ni was between 12 mg/m^3^ and 5 g/m^3^, which was the optimal culture concentration for methanogens [19]. The effect of Ni addition on anaerobic fermentation was mainly achieved by the action of Ni on cellulase activity and methanogenesis in different stages of fermentation [15].

The concurrence of different metals induced complex reactions and unpredictable results, which hindered the efficient utilization of biowaste. Lin et al. studied the degradation of volatile fatty acids (VFAs) by heavy metals and found that the addition of composite metals had a synergistic inhibitory effect on VFAs [20]. Ni showed synergistic effects with Cu, Mo, Co, and mercury (Hg), and antagonistic effects with Zn and Cd [20]. In 1985, Ahring et al. proposed that Ni could reduce the toxicity of Cd and Cu [21]. As one of the worldwide environmental and health concerned heavy metals, Cd was able to impact the composition and diversity of bacterial communities in aerobic or anaerobic biological sludges used for wastewater treatment [22]. Once accumulated in the biomass, Cd would enter the fermenter and impact the anaerobic fermentation process [23]. The presence of Cd would induce different performances of anaerobic fermentation with Fe and Ni in the feedstocks. However, to the best of our knowledge, studies on the impacts of the Cd together with Fe and Ni on the anaerobic fermentation process were lacking, which hindered the efficient utilization of biowastes.

This study explored the effects of Cd-containing heavy metals (Cd, Cd + Fe, and Cd + Ni) on biogas yields and composition, pH, oxidation reduction potential (ORP), NH_4_^+^–N, VFA components, and microbial communities during thermophilic anaerobic fermentation of cow manure and corn stover. The influences of Cd, Cd + Fe, and Cd + Ni on biogas production, process stability, and biodegradation, as well as microorganisms, were then obtained and compared. In particular, the VFAs’ generation and the relationship between VFA components and microorganisms were analyzed. This study is expected to lay the foundation for the application of heavy metal polluted biowaste in anaerobic fermentation.

## 2. Materials and Methods

### 2.1. Experimental Materials

Corn stover, the feedstock for anaerobic fermentation, was collected from the farmland in Tongzhou District, Beijing in November 2016. The harvested corn stover was air dried until moisture levels reached <10%. The dried stover was then ground into powder and passed through a 0.5 mm standard soil sieve.

Fresh cow manure was used as an inoculum, which was collected from the Yanqing base, Beijing Dairy Cattle Centre. The total solid (TS) of the raw cow manure was 16.49% ± 0.16% dry weight and the volatile solid (VS) was 84.00% ± 0.48% of TS. No extra inoculum was used to start the experiment. The properties of corn stover and cow manure were the same as in the previous study [24]. The Cd contents in the corn stover and cow manure were lower than the limit of detection and were elided in this study.

### 2.2. Anaerobic Fermentation Experiment

The experiments were performed in the anaerobic fermenters (total volume of 30 L, YGF 300/30, Shanghai Yangge Biological Engineering Equipment Co., Ltd., Shanghai, China) at 55.0 ± 1.0 °C for 28 days. The fermenters were autoclaved before each experiment. The working volume of the fermentation system was 20 L. A stirrer with three layers of mixing blades was set in the middle of the reactor to ensure complete mixing of the substrates.

Equal amounts of corn stover and cow manure were mixed as the substrates for fermentation, that is, 0.8 kg dry weight for each. The TS of the substrate in the reactors was adjusted to 8% by adding distilled water. At the beginning of fermentation, 0.041 g CdCl_2_·2½H_2_O was added into the fermenters with a final Cd concentration of 1.0 mg/L, which would not cause an inhibition according to the previous publication [23]. The group with only Cd addition was compared with experimental groups, adding extra 10.0 mg/L Fe (0.712 g FeCl_2_·4H_2_O addition) and 2.0 mg/L Ni (0.162 g NiCl_2_·6H_2_O addition), respectively. The selection of metal concentrations was based on the previous researches on individual metal addition [15,16,23], in which 1.0 mg/L initial Cd concentration, 10.0 mg/L initial Fe concentration, and 2.0 mg/L initial Ni concentration improved cumulative biogas yields. After adding metals, the fermenters were then infiltrated with high-purity nitrogen for 5 min to expel residual air.

### 2.3. Measurements

Biogas yields, pH values, and ORP were automatic measured at 09:00 every day [14]. TS was measured by weighing the samples after drying at 105 °C for 24 h. VS was measured after treating the samples in muffle for 550 °C, 1 h. Total nitrogen (TN) was measured using the Indophenol blue colorimetric method after being digested by concentrated sulfuric acid and 30% hydrogen peroxide [25]. Total organic carbon (TOC) was measured by the potassium dichromate volumetric method [26]. NH_4_^+^–N and VFA were measured every three days via the sampling port at the bottom of the reactor. Briefly, about 200 mL of sample was collected after fully stirring the substrate. NH_4_^+^–N concentrations in the supernatant were obtained by Nessler’s reagent [27] after centrifuging the sample at 5000 rpm for 10 min. Samples for VFA analysis were passed through a 0.45 μm nitrocellulose membrane filter and frozen prior to analysis. VFA concentrations were measured using a gas chromatograph (GC—2014, Shimadzu Co., Kyoto, Japan) with a flame ionization detector (FID). VFA was expressed as mg/L of individual species (C2–C5 fatty acids). CH_4_ contents in biogas were measured by a gas chromatograph (GC—2014C, Shimadzu Co., Kyoto, Japan) equipped with a GDX-401 column with H_2_ as the carrier gas. Detection was performed with a thermal conductivity detector (TCD).

The measurements of microbial communities were conducted by Novogene Co. Ltd (Beijing China) after certificating the samples at 8000 rpm and 4 °C for 3 min. Briefly, the genomic DNA of the samples on the 7th, 13th, and 19th day were extracted by the cetyltrimethylammonium ammonium bromide (CTAB) methods [28]. After the extraction, the samples were diluted to a concentration of 1 ng/L with sterile water. Then, the diluted genomic DNA was used as the template for PCR amplification. PCR amplification of the V3–V4 hypervariable region of bacterial 16S rDNA was performed using universal primers 338 F (50-ACTCCTACGGGAGGCAGCAG-30) and 806 R (50-GGACTACHVGGGTWTCTAAT-30) [29]. Archaea primers used to amplify the V3–V4 hypervariable region of archaeal 16S rDNA were 344 F (50-ACGGGGYGCAGCAGGCGCGA-30) and 806 R (50-GGACTACVSGGGTATCTAAT-30). All primers included Illumina barcode sequences for multiplexing each sample. The library construction was conducted with TruSeq^®^ DNA PCR-Free Sample Preparation Kit. After the Qubit and Q-PCR quantification, the constructed library was qualified and HiSeq2500 PE250 was used for sequencing.

### 2.4. Data Analysis

After removing the barcode and primers’ sequences, the reads were matched with FLASH (V1.2.7) for raw Tags and then sieved for clean Tags. Clean Tags were cut out and the length filtered by referencing the Qiime quality control process (V1.9.1). The obtained Tags were treated by removing the chimeric sequence through comparison with the detection chimeric sequence (Gold database), yielding the final effective Tags as the targets. The cluster analysis of effective Tags was conducted using Uparse software (V7.0.1001). The operational taxonomic units (OTUs) were clustered with identity >97%. Species annotation of the OTUs’ representative sequence was carried out using the Mothur method and Small subunit ribosomal RNA (SSUrRNA) database (define threshold of 0.8–1.0). The microbial communities were then obtained after annotation.

The data in the study were the average of three parallel repeats. Error bars represent the standard errors of the mean: SEM = SD/$\sqrt{n}$, where SD is the standard deviation. One-way analysis of variance (one-way ANOVA) and Pearson correlation analysis were performed in Statistical Package for the Social Science (SPSS, 17.0, Chicago, IL, USA) software at 0.05 and 0.01 levels of significance, represented by * (*p* < 0.05) and ** (*p* < 0.01), respectively.

## 3. Results and Discussion

### 3.1. Biogas Properties

#### 3.1.1. Cumulative and Daily Biogas Yields

Figure 1a shows the impact of Fe and Ni combined with Cd addition on cumulative biogas yields. The cumulative biogas yields of the Cd + Ni and Cd +Fe groups were enhanced by 119.76% and 80.89%, respectively, compared with the control group (only Cd added). The biogas yield of the Cd + Ni group far exceeded that of the control group in the first seven days. This was mainly because Ni was the active central component of the main enzyme involved in anaerobic fermentation and played a key role in the formation of CH_4_ by methyl coenzyme M reductase (MCR) [30,31]. Compared with the Cd group, the addition of Cd + Ni increased the activity of the enzyme. The result indicated that the addition of Cd + Ni was beneficial to start the anaerobic fermentation and usher in the peak of biogas yield in advance. The cumulative biogas yield of the Cd + Fe group surpassed that of the control group from the eighth day, which was partly because of the fact that Fe was the most abundant metal element in the tissue of methanogenic bacteria, and was necessary for the synthesis of various enzymes in the anaerobic fermentation (such as CO dehydrogenase, acetyl coenzyme A synthase, and hydrogenase), and could activate the activity of related enzymes [32]. Besides, the addition of Fe was able to reduce the concentration of dissolved sulfides during anaerobic fermentation, thus lowering the toxicity of sulfides to microorganisms [33]. After the 20th day, the cumulative biogas yield curve of the Cd and Cd + Fe groups reached a plateau, while that of the Cd + Ni group was still ascending.

The effects of Cd, Cd + Fe, and Cd + Ni addition on the daily biogas yields are shown in Figure 1b. The highest daily biogas yields were 15.56, 34.64, and 44.07 mL/g TS for the Cd, Cd + Fe, and Cd + Ni groups, respectively. The control group had two peaks of biogas yield throughout the fermentation. According to a previous study, the first biogas yield peak was caused by the degradation of cow manure, while the second biogas yield peak was the result of the degradation of corn stover [23]. Meanwhile, two peaks of the Cd + Fe group were only two days apart, and only one peak of the Cd + Ni group appeared. This indicated that the addition of Cd + Fe and Cd + Ni was beneficial to improve the hydrolysis efficiency of the non-degradable materials in the substrate and shorten the gap between the peaks of biogas yield. Thus, the additions of Fe and Ni into the Cd-containing anaerobic fermentation system were able to improve and bring forward the daily biogas peak. The above results showed that the addition of Cd + Fe and Cd + Ni improved the biogas yields of anaerobic fermentation compared with the control group, and the promotion effect of the Cd + Ni group on biogas yield was greater than that of the Cd + Fe group.

#### 3.1.2. CH_4_ Content

The percentage of CH_4_ in biogas was determined to compare the energy conversion efficiency and the impact of heavy metals addition. Figure 1c reports the CH_4_ content of the biogas in the Cd, Cd + Fe, and Cd + Ni groups. During the first four days of fermentation, the CH_4_ contents and the corresponding cumulative biogas yields in the Cd and Cd + Fe groups were not obviously different. The CH_4_ content of the Cd + Ni group was higher than that of the Cd + Fe and control groups, indicating the accelerating impact of Cd + Ni on the start-up of the fermentation. However, on the seventh day, the CH_4_ content in the Cd + Ni group suddenly decreased; on the contrary, the daily biogas yield reached the maximum. This could be because of the relatively low activity of methanogens at this time, which was caused by the sharp variation of pH values after the fifth day (Figure 2), and further affected the methanogenesis process of the Cd + Ni group. Along with the stability of the fermentation system and the adaption of microorganisms, the hydrolytic products were used by methanogens, resulting in an increase in CH_4_ content on the 10th day. The highest CH_4_ contents were 67.77%, 73.17%, and 80.76% for the Cd, Cd + Fe, and Cd +Ni groups, respectively. The CH_4_ content in the Cd + Ni group was greater than that of the Cd and Cd + Fe groups. The CH_4_ content in the Cd + Ni group in this study was higher than in the previous report (70.41%) when only 2.0 mg/L Ni was added to reed and cow dung [15]. Similarly, the maximum CH_4_ content in Cd + Fe in this study was higher than that of CH_4_ content (45.33%) adding 10.0 mg/L Fe alone in a previous study [16]. Therefore, the additions of Cd + Ni and Cd + Fe were beneficial to increase the content of CH_4_ in the fermentation process and improve the quality of biogas.

### 3.2. Process Stability

#### 3.2.1. pH Values

The importance of pH values on representing the status of fermenters and impacting the activity of microbial communities has been widely considered [34,35]. The variations of pH values during the fermentation process are shown in Figure 2a. Overall, the pH of the Cd + Fe and Cd + Ni groups increased in the first few days of fermentation, then decreased, and finally rose to a stable state, while the pH of the control group was always unstable. In general, during the start-up stage of fermentation, organic substances in the substrate were rapidly hydrolyzed to acid, which led to the accumulation of acidic hydrolysates, resulting in lower pH value and lower biogas production. In the latter stage, the pH increased and stabilized through the utilization of VFA by methanogens [14]. However, the results in this study were different from this situation. At the beginning of fermentation (first two days), the pH of all groups increased. On one hand, the acid-producing microorganism in the fermentation system might not be adapted to the environment, resulting in less acid components in anaerobic fermentation. On the other hand, the increase in NH_4_^+^–N concentration (Figure 3) could buffer the acid components produced. In addition, there may be other acid–base substances in anaerobic fermentation that affected the increase of pH values. With the progress of anaerobic fermentation, the activity of acid-producing microorganisms was enhanced, and the acid-producing process was the rate-limiting step [36]. The pH of all groups in the anaerobic fermentation decreased. In the middle stage of fermentation, the CH_4_-producing stage dominated and the acid components generated in the acid producing stage were consumed, so the pH increased and eventually tended to be stable [36]. With the development of fermentation, the pH of the control group decreased for a long time (third to eighth day) as a result of the accumulation of acid components in the acid production stage. The pH of the Cd + Ni group and the Cd + Fe group decreased for a short time and increased rapidly, which might be because of the synergistic effects of the heavy metals in the process of CH_4_ production, and the more efficient utilization of acid components by methanogens [31]. Therefore, the pH of the Cd + Ni and Cd + Fe groups did not decrease remarkably, and the average pH of the whole fermentation process was 8.82% and 7.77% higher, respectively, than the control group.

During the whole fermentation process, the average pH values of the Cd, Cd + Fe, and Cd + Ni groups were 6.69 ± 0.12, 7.28 ± 0.15, and 7.21 ± 0.21, respectively, which were all located in the pH range suitable for anaerobic microorganisms [37]. From the 5th day to 13th day, the increase of pH toward the neutral condition in the Cd + Ni and Cd + Fe group corresponded with the achievement of the peak biogas stage. After the 13th day, the pH of the Cd + Fe and Cd + Ni groups began to stabilize, while the pH of the control group changed remarkably. This result indicated that the additions of Cd + Fe and Cd + Ni were beneficial to improve the stability of the fermentation process and enhance the acid–base buffering capacity of the fermentation system.

#### 3.2.2. Oxidation Reduction Potential (ORP)

The ORP value represented the oxidation reduction reactions in the fermenters and was a useful parameter to monitor the fermentation process [38]. The effects of Cd, Cd + Fe, and Cd + Ni on ORP values during fermentation are shown in Figure 2b. On the first day, the ORP values of the groups were in the order of Cd + Ni < Cd + Fe < Cd. From the second to the ninth day of fermentation, the ORP values of the Cd and Cd + Fe groups decreased and fluctuated, while the ORP value of the Cd + Ni group was higher and stable. The results showed that the redox state of anaerobic fermentation in the Cd + Ni group was placid. During this period, the daily biogas yield and CH_4_ content of the Cd + Ni group were higher than those of the other groups. The ORP value of the Cd + Fe group remained relatively stable, while the ORP value of the control group increased starting from the 10th day of fermentation.

### 3.3. Substrate Biodegradation

#### 3.3.1. NH_4_^+^–N Concentration

NH_4_^+^–N was one of the sources of nutrient elements required by anaerobic microorganisms. An appropriate amount of NH_4_^+^–N could promote the activity of methanogens [39]. The methanogens, as members of the archaea, were among the microbial populations most sensitive to NH_4_^+^–N in anaerobic fermentation [40]. In addition, NH_4_^+^–N affected the activity of CH_4_ synthase. The effects of Cd, Cd + Fe, and Cd + Ni on the NH_4_^+^–N concentrations in the fermentation are shown in Figure 3. Overall, the NH_4_^+^–N concentrations in all of the groups followed the same trend of an increase first and then a decrease. However, there was still a big difference between the compound metal added groups and the control group. The NH_4_^+^–N concentration on the fourth day was highest in the Cd + Ni group because of the addition of Ni contributing to the release of the nitrogen source. After the 10th day, the concentrations of NH_4_^+^–N in the control group varied greatly, while the concentrations of NH_4_^+^–N in the compound metals groups were relatively stable. During this period, the order of NH_4_^+^–N concentration was Cd + Ni > Cd + Fe > Cd, which was same as the order of cumulative biogas yield.

NH_4_^+^–N was reported to promote the production of CH_4_ when the concentration was <6000 mg/L, while NH_4_^+^–N was able to increase the alkalinity of the fermentation system and the buffering of VFA [41]. High NH_4_^+^–N concentrations brought high free ammonia nitrogen, which has been a main cause of inhibition owing to its high permeability to the bacterial cell membrane. Ammonia might affect methanogens in two ways: (i) ammonium ion inhibited the CH_4_ producing enzymes directly, and/or (ii) hydrophobic ammonia molecule diffused passively into bacterial cells, causing proton imbalance or potassium deficiency [42]. In the present study, the peak of NH_4_^+^–N concentration of all groups was far below 6000 mg/L. Therefore, the addition of Cd together with Fe and Ni enhanced the stability of the NH_4_^+^–N concentration, which in turn helped to buffer the pH of the fermentation solution, making the fermentation system more suitable for producing CH_4_.

#### 3.3.2. Responses of VFA

The concentrations of total VFA and the compositions, that is, acetic acid, propionic acid, butyric acid, valeric acid, and isovaleric acid, are presented in Figure 4. Overall, the VFA concentrations of all the groups in the initial stage were higher than in the middle and late stage of the fermentation. During the first 10 days of fermentation, total VFA concentrations of the Cd + Fe group were higher than those of the Cd and Cd + Ni groups, with the peak concentration of total VFA on the seventh day. During this period, despite the high total VFA concentrations, the pH values did not declined because alkaline components were generated as well, such as NH_4_^+^–N (Figure 3). The VFA concentration in the Cd + Ni group increased from the first to the fourth day, as did the daily biogas yields. On the seventh day, the increases of VFA in the Cd and Cd + Fe groups were observed together with the increment of daily biogas yields. The synchronous increases of total VFA and daily biogas yields suggested that the VFA was efficiently generated and consumed for biogas production. On the 13th day, the VFA concentration in the Cd + Fe group decreased sharply, while the daily biogas yield increased to a maximum. This result indicated that the rate of VFA consumed by biogas production was higher than the rate of VFA produced.

The main compositions of VFA varied with metal additions. During the first seven days, acetic acid was the main composition of VFA, followed by butyric acid > propionic acid > valeric acid in the Cd and Cd + Fe groups. However, the order of the VFA compositions was butyric acid > propionic acid > acetic acid > valeric acid in the Cd + Ni group. Among all VFAs, acetic acid and butyric acid were the most favorable for CH_4_ formation, while contribution of acetic acid was more than 70% [43]. Thus, acetic acid was efficiently used for the methanogenesis in the Cd + Ni group, which contributed to the higher biogas and CH_4_ yields. During the anaerobic fermentation process, the average VFA concentration in the Cd + Ni group was lower than that in the Cd + Fe group, indicating that the VFA consumption rate in the Cd + Ni group was higher than that in the Cd + Fe group. Meanwhile, Ni was previously proved to be the most effective element compared with Fe, Co, and Mo, exhibiting the maximum increment of biogas yield [30]. Therefore, the biogas yield of the Cd + Ni group was higher than that of the Cd + Fe group. After the 10th day, the concentrations of VFA compositions in all groups decreased.

Figure 5 shows the Pearson correlation between pH values and VFA concentrations. There were significantly negative correlations between pH values and VFA concentrations in the Cd + Fe (−0.781, *p* < 0.01) and Cd + Ni (−0.875, *p* < 0.01) groups. Previous studies showed that the increase and accumulation of VFA concentrations resulted in a corresponding decrease in the pH value of suspension, and both VFA concentrations and the pH values of suspension affected the activity of hydrolytic microorganisms in an anaerobic environment [44]. At the beginning of fermentation, the organic matter in the substrate was rapidly hydrolyzed to acid, such as VFA, which cannot be effectively used for methanogenesis owing to the slow adaptation and metabolism of methanogens. Therefore, the accumulation of acidic hydrolyzed products led to a decrease in pH and a low biogas yield [45]. In the control group, pH values were not correlated to VFA concentrations. This was probably because of the influence of other acid–base substances on pH in the control group, which needed further study and analysis.

### 3.4. Microbial Properties

#### 3.4.1. Structure of Bacterial Communities

The variations of bacterial communities annotated on the level of genus are shown in Figure 6. It was found that the compound metals exerted different influences on bacterial communities as the fermentation progressed. The changes in bacterial community mainly manifested in the relative abundance of the same bacterial genus at different fermentation stages. On the seventh day, the dominant bacteria of the Cd addition group were recognized as *Ruminiclostridium*, *Mobilitalea*, *Tepidimicrobium*, and *Ruminiclostridium_1*. As previously reported by Fosses et al., *Ruminiclostridium cellulolyticum* was able to produce extracellular multi enzymatic complexes called cellulosomes, which efficiently degraded the crystalline cellulose and the cell wall [46]. *Mobilitalea* was reported as a novel strictly anaerobic, halotolerant, organotrophic bacterium, strain P3M-3T. Strain P3M-3T grew optimally at 37 °C, pH 7.0–7.5, and in an NaCl concentration of 15 g/L [47]. Under optimum growth conditions, the doubling time was 1 h. *Mobilitalea* was able to ferment a variety of mono-, di-, and polysaccharides, including microcrystalline cellulose [47]. *Tepidimicrobium* was anaerobic, moderately thermophilic and neutrophilic, with the temperature range for growth of 25–67 °C, and pH range for growth of 5.5–9.5 [48]. *Tepidimicrobium* spp. grew organotrophically on a number of proteinaceous substrates, amino acids, and carbohydrates and produced acid, ethanol, H_2_, and CO_2_ [48]. Therefore, the detected bacterial communities in the Cd addition group supported the degradation of carbohydrates and proteins as well as the generation of CH_4_.

In the Cd + Fe group, the dominant bacteria were *Ruminiclostridium*, *Tepidimicrobium*, *Caproiciproducens*, and *Ruminiclostridium_1* on the seventh day. *Caproiciproducens* was a strictly anaerobic, Gram-stain-positive, non-spore-forming, rod-shaped bacterial strain [49]. They grew at 35–45 °C and pH of 6.0–8.0, and produced H_2_, CO_2_, ethanol, acetic acid, butyric acid, and caproic acid as metabolic end products of anaerobic fermentation [49]. *Ruminiclostridium* and *Ruminiclostridium_1* were increased on the 13th and 19th day alongside the reduction of *Tepidimicrobium* and *Caproiciproducen*. The result suggested that, as the fermentation progressed, the substrate was continuously consumed, resulting in a decreased relative abundance of *Tepidimicrobium* and *Caproiciproducen*.

In the Cd + Ni group, the relative abundance of *Ruminiclostridium_1* was the highest, followed by *Tepidimicrobium*, *Defluviitoga*, and *Mobilitalea*, on the seventh day. A new isolate L3 of *Defluviitoga tunisiensis* presumably was able to degrade cellulose, as genes encoding cellulases were identified in its genome [50]. Acetate, H_2_, and CO_2_ were supposed to be end products of the fermentation process. The relative abundance of *Defluviitoga* was highest on the 19th day. Therefore, H_2_ and CO_2_ were efficiently generated and used for CH_4_ production at this time.

#### 3.4.2. Methanogens and Their Relationships with Fermentation Parameters

According to the results of clustering and species annotation of OTUs, the absolute abundance in the fermentation system at the archaeal genus level on the 7th, 13th, and 19th was analyzed to focus on the variation of methanogens. Figure 7 shows the effect of Cd, Cd + Fe, and Cd + Ni addition on the absolute abundance of seven methanogens, namely *Methanothermobacter*, *Methanobrevibacter*, *Methanobacterium*, *Methanosphaera*, *Methanocorpusculum*, and *Candidatus_Methanoplasma*.

It can be seen from Figure 7 that the total abundance of methanogens in the Cd added group on the seventh day was much higher than that of the other groups with *Methanobrevibacter* as the dominant genus. All of the *Methanobrevibacter* species are hydrogenotrophs [51] and use H_2_ and formate as substrate for their CH_4_ production [52]. However, the *Methanobrevibacter* in the Cd + Ni group was not of a large amount, which was probably caused by the low concentrations and bioavailability of Ni. The total abundance of methanogens in the Cd + Fe group on the 13th day was higher than that of the other groups with *Methansobrevibacter* as the dominant genus, followed by *Methanobacterium*. The total abundance of methanogens in the Cd + Ni group was higher than that of the other groups on the 19th day. The main methanogens in this group on the 19th day were identified as *Methanothermobacter* and *Methanobacterium*.

The relationships between the methanogens and fermentation parameters according to the Pearson correlation analysis are shown in Figure 8. The vertical direction is the information of different environmental factors, and the horizontal direction is the representation of the methanogens’ information. In the Cd added group, *Methanobrevibacter* (*p* < 0.01) and *Methanobacterium* (*p* < 0.05) were negatively correlated to pH. In the Cd + Fe group, *Methanothermobacter* was positively correlated to pH (*p* < 0.05), but negatively correlated to NH_4_^+^–N (*p* < 0.05) and acetic acid (*p* < 0.01). Besides, *Methanobrevibacter* (*p* < 0.05) and *Methanobacterium* (*p* < 0.05) were positively correlated to daily biogas yields in the presence of Fe. There was no significant relationship between the methanogens and fermentation parameters in the Cd + Ni group.

As can be seen from the above results, there were obvious bacterial community structure changes in different fermentation stages of all compound metals groups. *Ruminiclostridium*, *Mobilitalea*, *Tepidimicrobium Ruminiclostridium_1*, *Caproiciproducens*, and *Defluviitoga* were the dominant genus under all tested conditions, and the relative abundance between them varied with the fermentation progress. On the basis of the relationships between the methanogens and fermentation parameters, it can be concluded that different microbial strains responded differently to environmental factors. The methanogens during the fermentation process were impacted by pH values in the presence of Cd. After the addition of Fe, the relationship between methanogens and pH was modified by the impacts of Fe on NH_4_^+^–N and acidic components. In the presence of Fe, both *Methanobrevibacter* and *Methanobacterium* were found to be positively correlated with the daily biogas yields, suggesting that Fe impacted the biogas production by affecting the methanogens. For the Cd and Cd + Ni groups, no methanogens were detected to be related with biogas production, which may be because of different environmental factors; further research is needed.

According to Figure 8, there was no significant correlation between methanogens and total VFA concentrations. Figure 9 shows the correlation analysis between VFA components and methanogens under different conditions. In the Cd + Fe group, *Methanothermobacter* showed significant negative correlation with acetic acid (*p* < 0.01), butyric acid (*p* < 0.01), and valeric acid (*p* < 0.05). A possible reason for this result was that increases in the acetic acid, butyric acid, and valeric acid advanced acid bacteria growth, and consequently accelerated the conversion to acetic acid, followed by decreases in the activity of methanogens. These findings were consistent with the results of a previous study [53]. For the Cd and Cd + Ni groups, no methanogens were detected to be related with VFA components, which may be the result of different operating conditions. Further research is needed.

## 4. Conclusions

This research studied the effects of Cd combined with Fe and Ni separately on anaerobic fermentation with mixed corn stover and cow manure as feedstocks. The addition of compound metals improved the biogas yields and CH_4_ contents remarkably. Cd + Fe and Cd + Ni addition led to better process stability and improved the substrate biodegradation of fermentation, which enhanced the stability of NH_4_^+^–N during the fermentation process and brought forward the consumption of VFA. The structure of bacterial communities and the abundance of methanogens were influenced by the addition of compound metals. Adding Fe into the Cd-containing system resulted in a different relationship between methanogens and environmental parameters, in particular pH values, and induced higher biogas yields. The suggestion for future research and practice is developing a mixed microbial agent for improving the degradability and CH_4_ yields. 

## Figures and Tables

**Figure 1 ijerph-16-04066-f001:**
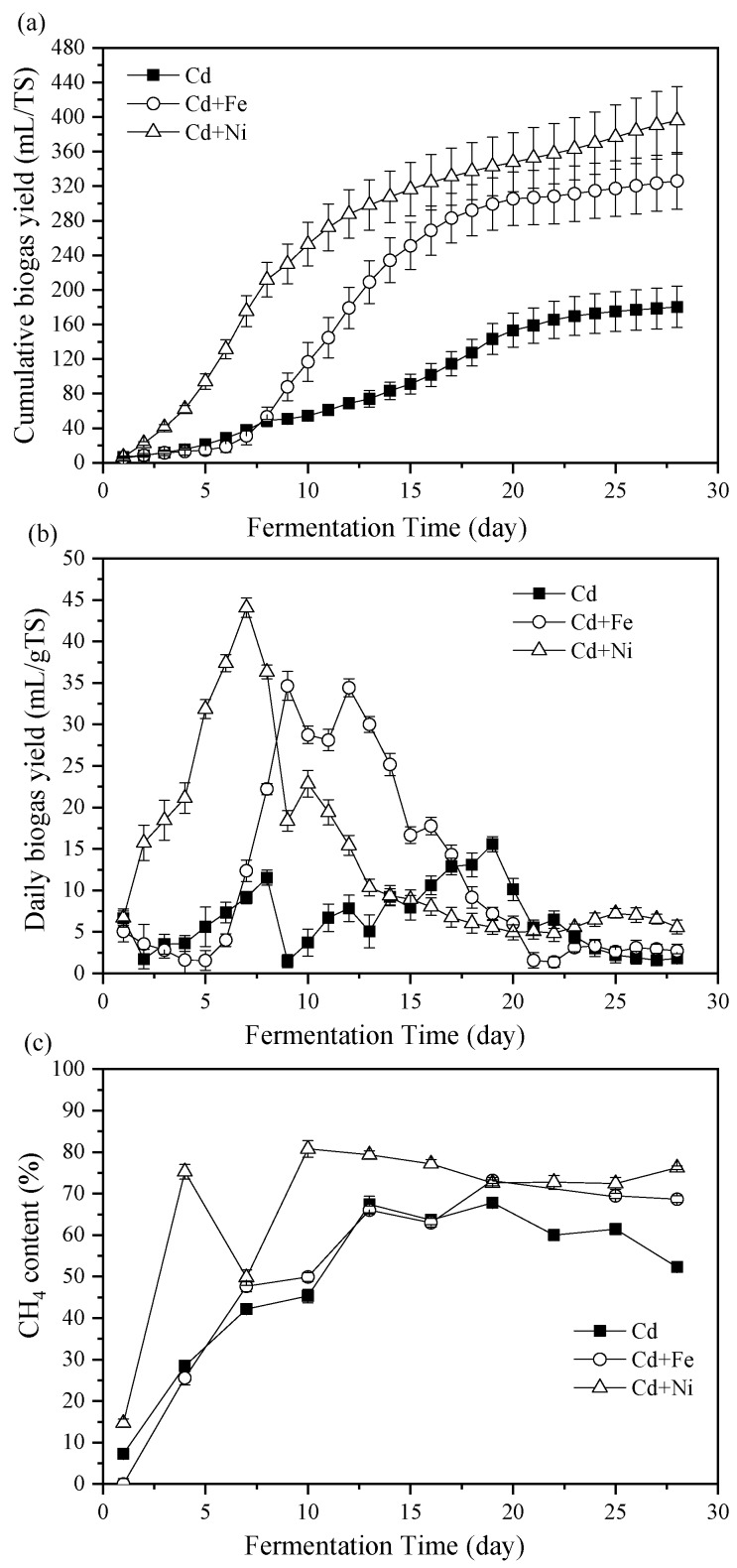
Cumulative biogas yields (**a**), daily biogas yield (**b**), and CH_4_ contents (**c**) in response to Cd, Cd + Fe, and Cd + Ni addition during the fermentation.

**Figure 2 ijerph-16-04066-f002:**
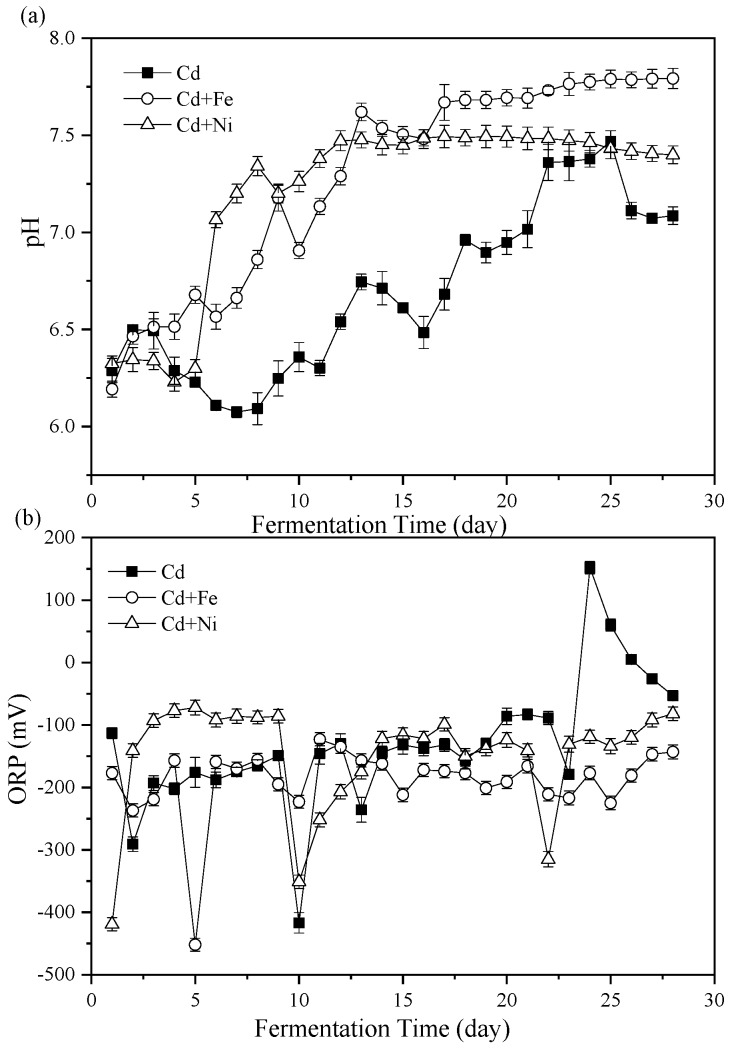
Impact of Cd, Cd + Fe, and Cd + Ni addition on pH values (**a**) and oxidation reduction potential (ORP) (**b**) during the fermentation.

**Figure 3 ijerph-16-04066-f003:**
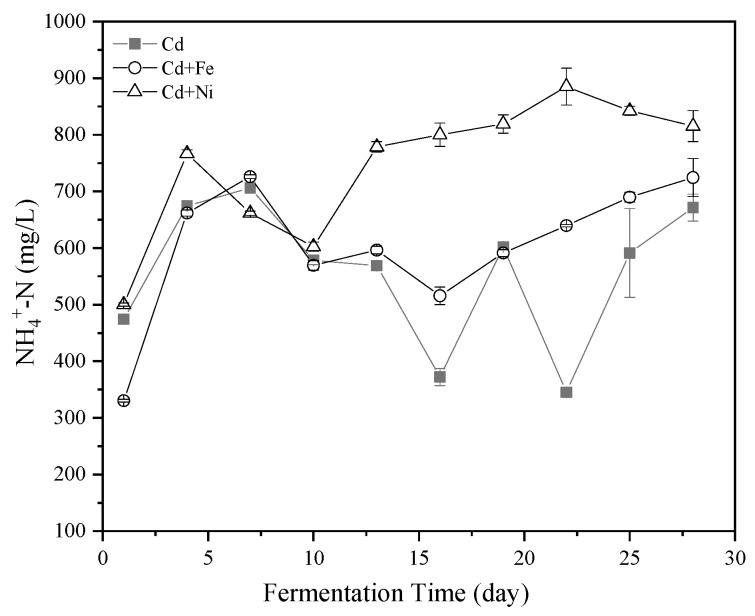
Impact of Cd, Cd + Fe, and Cd + Ni addition on NH_4_^+^–N concentrations during the fermentation.

**Figure 4 ijerph-16-04066-f004:**
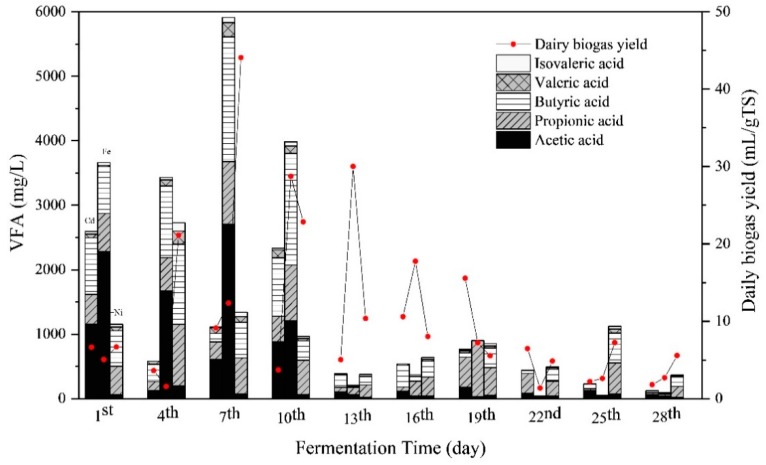
Impact of Cd, Cd + Fe, and Cd + Ni addition on volatile fatty acids (VFAs) and daily biogas yield during the fermentation.

**Figure 5 ijerph-16-04066-f005:**
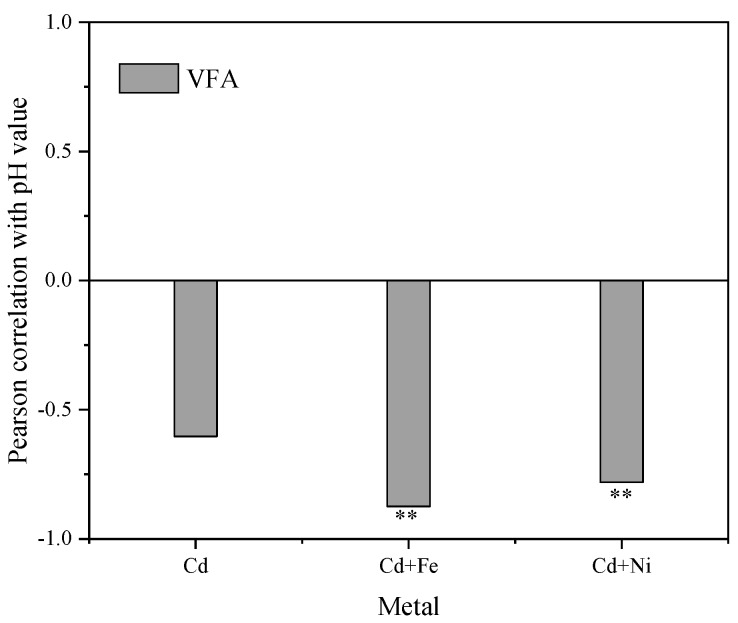
Relationships between pH values and VFA concentrations during the fermentation. **, *p* < 0.01.

**Figure 6 ijerph-16-04066-f006:**
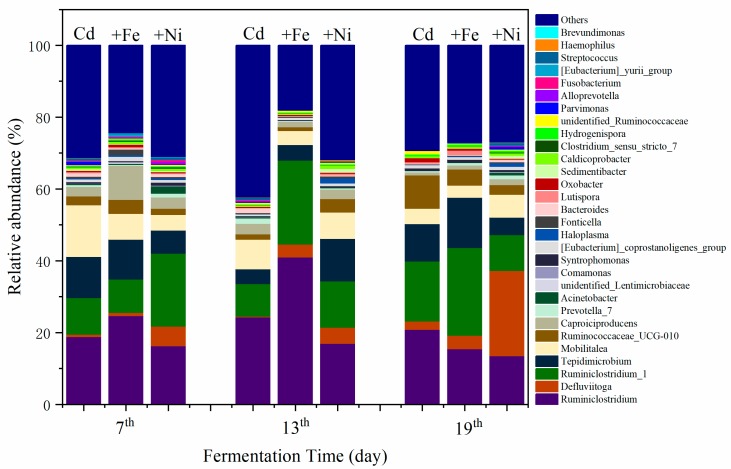
Impact of Cd, Cd + Fe, and Cd + Ni addition on the structure of bacterial communities during the fermentation.

**Figure 7 ijerph-16-04066-f007:**
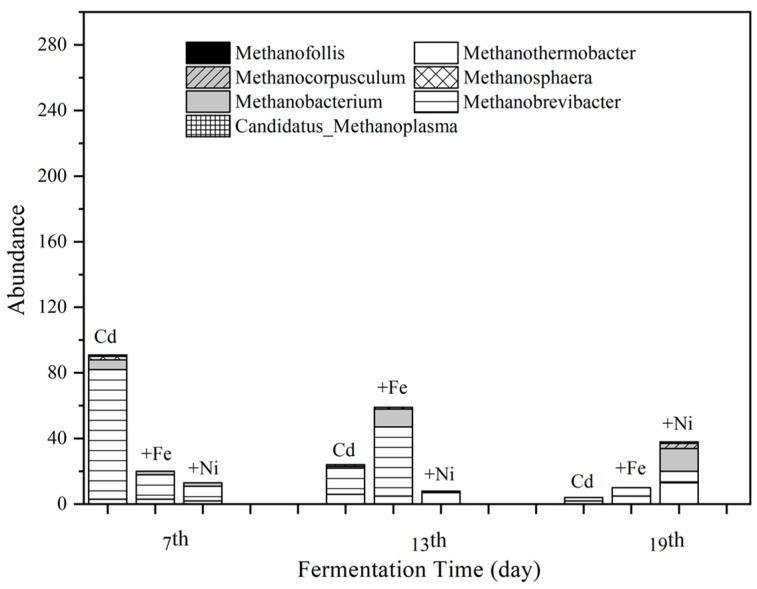
Impact of Cd, Cd + Fe, and Cd + Ni addition on the absolute abundance of methanogens during the fermentation.

**Figure 8 ijerph-16-04066-f008:**
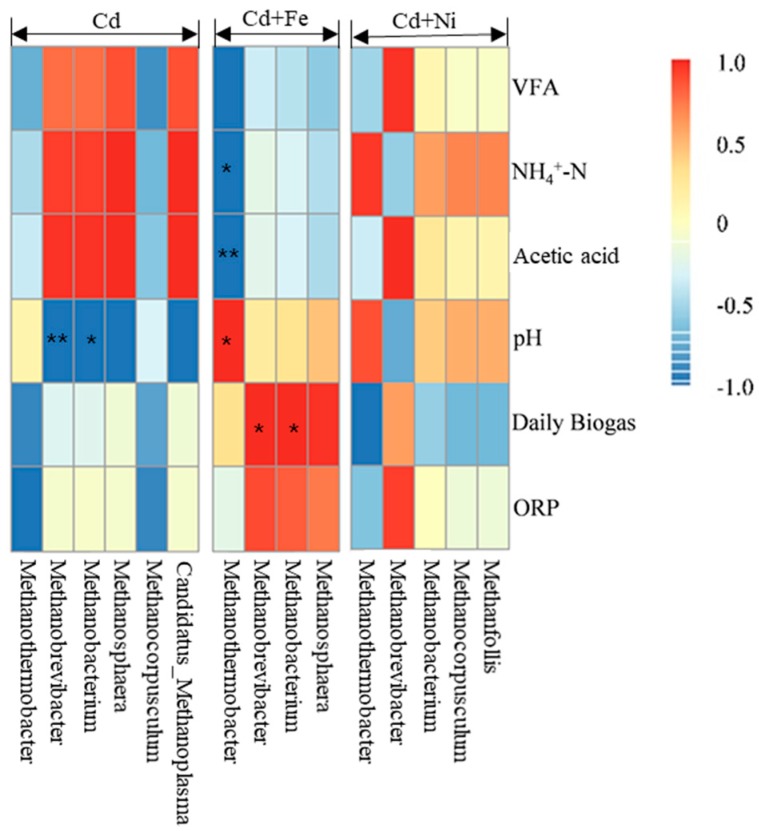
Pearson correlation analysis between environmental factors and methanogen under Cd, Cd + Fe, and Cd + Ni addition during the fermentation.

**Figure 9 ijerph-16-04066-f009:**
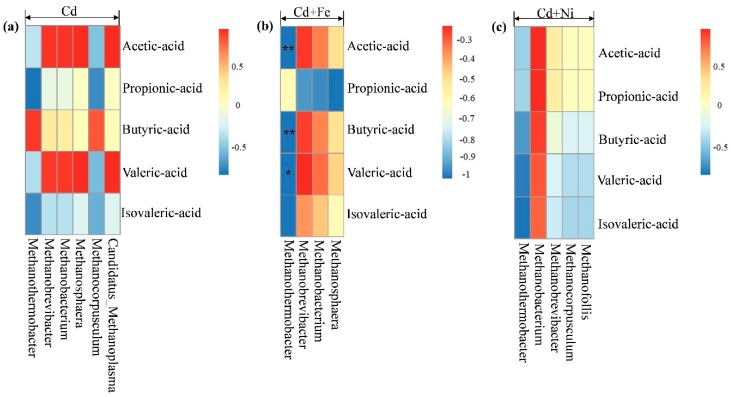
Pearson correlation analysis between VFA components and methanogen under Cd (**a**), Cd + Fe (**b**), and Cd + Ni (**c**) addition during the fermentation.

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
