# Peer review of "Impact of Fe and Ni Addition on the VFAs’ Generation and Process Stability of Anaerobic Fermentation Containing Cd"

_ijerph, 2019, doi:10.3390/ijerph16214066_

Round 1
Reviewer 1 Report
The manuscript can benefit by addressing the following issues/concerns.
The writing must be improved. Please relate pH with VFA concentration in the discussion section. The main issue with the study is on the design of experiment or the lack thereof. It seems that Cd, Fe, and Ni were added in concentrations based on literature. No exploratory initial experiment at all. With respect to application of study, it becomes a question of whether the results can be applied to other waste stream. Is the ratio of Cd:Fe:Ni at the concentrations used in this study applicable to any TS, or VS, or C/N. Or should the concentration be adjusted depending on initial TS/VS of substrate? The next issue is data reliability. The study was conducted with no replicates. This becomes an issue when the impact of something is being evaluated. Since there are no standard deviations, there is no way to tell whether the impact of something is significant or not.Author Response
Dear reviewer,
We really appreciate your useful comments and suggestions on our manuscript. According to the comments and suggestions, we have checked and modified the manuscript carefully. The detailed corrections are listed below. We have marked the changes in the revised manuscript in red color. We highly appreciate your attention and consideration on our work.
Sincerely,
Dr. Huayong Zhang
Point 1: Please relate pH with VFA concentration in the discussion section.
Response 1: Thank you very much for your suggestions. We added the relationships between pH values and VFA concentrations during the fermentation in Responses of VFA section according to the comments and marked in the revised manuscript.
Point 2: The main issue with the study is on the design of experiment or the lack thereof. It seems that Cd, Fe, and Ni were added in concentrations based on literature. No exploratory initial experiment at all.
Response 2: Thank you very much for your suggestions. The present experiments were designed basing on our previous studies on investigating the performance of anaerobic fermentation with Cd, Fe and Ni, respectively [1,2,3]. The results showed that 1 mg/L initial Cd concentration, 10 mg/L initial Fe concentration and 2.0 mg/L initial Ni concentration improved cumulative biogas yields relative to other concentrations. Therefore, we supposed that the data shown in the present study would support for the effect of Cd, Cd + Fe and Cd + Ni addition on anaerobic fermentation.
Point 3: With respect to application of study, it becomes a question of whether the results can be applied to other waste stream. Is the ratio of Cd: Fe: Ni at the concentrations used in this study applicable to any TS, or VS, or C/N. Or should the concentration be adjusted depending on initial TS/VS of substrate?
Response 3: Thank you very much for your suggestions. For waste stream with similar properties to the materials used in present study, the ratio of Cd: Fe: Ni in present study can be used as a reference. The concentration of heavy metals in the fermenter should be adjusted according to the TS of the raw materials.
Point 4: The next issue is data reliability. The study was conducted with no replicates. This becomes an issue when the impact of something is being evaluated. Since there are no standard deviations, there is no way to tell whether the impact of something is significant or not.
Response 4: Thank you very much for your suggestions. The figures have been revised. Standard deviations in Figure 1, Figure 2 and Figure 3 have been added into the revised manuscript.
Reference
Tian Y.L.; Zhang H.Y.; Chai Y. Biogas properties and enzymatic analysis during anaerobic fermentation of Phragmites australis straw and cow dung: influence of nickel chloride supplement. Biodegradation 2017, 28, 15-25. Zhang H.Y.; Tian Y.L.; Wang L.J. Effect of ferrous chloride on biogas production and enzymatic activities during anaerobic fermentation of cow dung and Phragmites straw. Biodegradation 2016, 27, 69-82. Zhang H.Y; Tian Y.L.; Wang L.J. Ecophysiological characteristics and biogas production of cadmium-contaminated crops. Bioresour. Technol. 2013, 146, 628-636.
Reviewer 2 Report
I consider that the study performed and presented in the manuscript “Impact of Fe and Ni addition on the VFAs generation and process stability of anaerobic fermentation containing Cd” is of interest for audience looking at the vaporization of waste biomass and energy production. Thus, I believe that this article is suitable for the International Journal of Environmental Research and Public Health. However, in some sections the presentation of the results and discussion is to descriptive and a major effort in giving answer to the results obtained is required. I have the following observations:
4 L164: You have reported the improvement in cumulative and daily bio-gas by the addition of Cd + Fe and Cd+Ni when compared only to Cd addition. Can you please provide a more in depth discussion to this result? 4 L16: Why does the combination Cd+Ni have a greater effect in biogas production that Cd+Fe?
5, L181; Figure 1(c): In the profile of CH4 production with the addition of Cd+Fe there is a drop around the 7th day then by the 10th day values raise again. Please explain what is causing this drop and why does the CH4 % value reach the same % value as in time 4 d.
5, L194: Please explain why is the Cd+Ni addition give grater % of methane that Cd+Fe.
6, L209: Why is there a pH increase during biogas production. Moreover, why does Cd+Fe and Cd+Ni addition have a greater effect on the increments of pH values?
8, L246: It is clear that Cd+Fe and Cd+Ni enhanced the stability of NH4+, but what is the relation of this with the metabolism of the microbiota present in the fermentation.
Author Response
Dear reviewer,
We really appreciate your useful comments and suggestions on our manuscript. According to the comments and suggestions, we have checked and modified the manuscript carefully. The detailed corrections are listed below. We have marked the changes in the revised manuscript in red color. We highly appreciate your attention and consideration on our work.
Sincerely,
Dr. Huayong Zhang
Point 1: 4 L164: You have reported the improvement in cumulative and daily bio-gas by the addition of Cd + Fe and Cd + Ni when compared only to Cd addition. Can you please provide a more in depth discussion to this result? 4 L16: Why does the combination Cd + Ni have a greater effect in biogas production that Cd + Fe?
Response 1: Thank you very much for your suggestions. We modified Cumulative and daily biogas yields section according to the comments and marked in the revised manuscript.
(a) A more in-depth discussion about the improvement in cumulative and daily biogas yields by the addition of Cd + Fe and Cd + Ni when compared only to Cd has been added in Cumulative and daily biogas yields section
(b) According to the results, within the first 10 days of anaerobic fermentation, VFA concentration in Cd + Ni group was lower than that in Cd + Fe group (Figure 4), indicating that VFA consumption of Cd + Ni group was more efficient than that of Cd + Fe group. Besides, Ni was previously proved to be the most effective element compared to Fe, Co and Mo, exhibiting the maximum increment of biogas yield [1]. Therefore, the biogas yield of Cd + Ni group was higher than Cd +Fe group.
Point 2: 5, L181; Figure 1(c): In the profile of CH4 production with the addition of Cd + Fe there is a drop around the 7th day then by the 10th day values raise again. Please explain what is causing this drop and why does the CH4 % value reach the same % value as in time 4 d.
Response 2: Thank you very much for your comments. We supplemented some important remarks in CH4 content according to the suggestion and marked in the revised manuscript.
(a) On the 7th day, the CH4 content in group Cd + Ni suddenly decreased, on the contrary the daily biogas yield reached the maximum. This could be due to the relatively low activity of methanogens at this time (Figure 7), which was caused by the sharp increase of pH values after the 5th day (Figure 2), and further affected the methanogenesis process of Cd + Ni group as well as the CH4 content.
(b) Along with the stability of fermentation system and the adaption of microorganisms, the hydrolytic products were used by methanogens resulting in an increase in CH4 content on the 10th day.
Point 3: 5, L194: Please explain why is the Cd+Ni addition give grater % of methane that Cd+Fe.
Response 3: Thank you very much for your suggestions. The CH4 content in Cd + Ni group was greater than Cd and Cd + Fe groups, because that Ni participated in the synthesis of a variety of important enzymes in the process of CH4 production [2], which promoted the utilization of intermediate products, such as VFA, by methanogens (manifested as the overall VFA concentration of Cd + Ni group was lower than that of Cd + Fe group, Figure 4), thereby increasing the CH4 production.
Point 4: 6, L209: Why is there a pH increase during biogas production. Moreover, why does Cd + Fe and Cd + Ni addition have a greater effect on the increments of pH values?
Response 4: Thank you very much for your comments. We modified the pH values section according to the comments and marked in the revised manuscript.
(a) At the beginning of fermentation (first 2 days), pH of all groups increased. On one hand, the acid-producing microorganism in the fermentation system might not be adapted to the environment, resulting in less acid components in anaerobic fermentation. On the other hand, the increase in NH4+-N concentration (Figure 3) could buffer the acid components produced. In addition, there may be other acid-base substances in anaerobic fermentation that affected the increase of pH. With the progress of anaerobic fermentation, the activity of acid-producing microorganisms was enhanced, and the acid-producing process was the rate-limiting step [3]. The pH of all groups in the anaerobic fermentation decreased. In the middle stage of fermentation, the CH4-producing stage dominated and the acid components generated in the acid producing stage were consumed [3], so the pH increased and eventually tended to be stable.
(b) With the development of fermentation, the pH of the control group decreased for a long time (3rd to 8th day) due to the accumulation of acid components in the acid production stage. The pH of Cd + Ni group and Cd + Fe group decreased for a short time and increased rapidly, that might be due to the synergistic effects of the heavy metals in the process of CH4 production, and the more efficient utilization of acid components by methanogens [2].
Point 5: 8, L246: It is clear that Cd + Fe and Cd + Ni enhanced the stability of NH4+-N, but what is the relation of this with the metabolism of the microbiota present in the fermentation.
Response 5: Thank you very much for your comments. Appropriate amount of NH4+-N could promote the activity of methanogens. In addition, NH4+-N affected the activity of CH4 synthase [4]. The methanogens as members of the archaea were among the microbial populations most sensitive to NH4+-N in anaerobic digestion [5]. High NH4+–N concentrations brought high free ammonia nitrogen, which had been considered to be a main cause of inhibition due to its high permeability to bacterial cell membrane. Ammonia might affect methanogens in two ways: (i) ammonium ion inhibited the CH4 producing enzymes directly and/or (ii) hydrophobic ammonia molecule diffused passively into bacterial cells, causing proton imbalance or potassium deficiency [6]. The text in the manuscript was modified correspondingly.
References
Zhang W.; Lei Z.; Li A. Enhanced anaerobic digestion of food waste by trace metal elements supplementation and reduced metals dosage by green chelating agent [S, S]-EDDS via improving metals bioavailability. Water Res 2015, 84, 266-277. Tang M.; Xian P.; Ying X U. Effects of Fe2+, Co2+ and Ni2+ on anaerobic process for landfill leachate treatment. Acta Scientiae Circumstantiae 2014, 34, 2573-2579. Zhang X.; Qiu W.; Chen H. Enhancing the hydrolysis and acidification of steam-exploded cornstalks by intermittent pH adjustment with an enriched microbial community. Bioresour. Technol 2012, 123, 30-35. Liu R.; Wang Y.; Sun C. Experimental study on biogas production from vegetable waste by anaerobic fermentation. Trans. CSAE 2008, 24, 209-213. Zhang Y.; Cañas M.Z.; Zhu Z. Robustness of archaeal populations in anaerobic co-digestion of dairy and poultry wastes. Bioresour. Technol 2011, 102, 779-785. Liu Z.; Dang Y.; Li C. Inhibitory effect of high NH4+–N concentration on anaerobic biotreatment of fresh leachate from a municipal solid waste incineration plant. Waste Manag 2015, 43, 188-195.
Reviewer 3 Report
September 26th, 2019
This manuscript presents that the effects of Cd combining with Fe and Ni on anaerobic fermentation with corn stover combined with cow manure. This would be an interesting manuscript for readers of International Journal of Environmental Research and Public Health Journal, but this study is very similar to the authors’ previous journal published August 2019, titled “Effect of Cd and Zn mixture on the biodegradation and microbial communities of anaerobic fermentation process.” The reviewer believes that the concept, methods, technical approach, many expressions of the sentences and paragraphs in the previous manuscript and current works are overlapped that dilute the novelty of the work.
For example, the results of table 1 is almost same with few changes in the words. This is not acceptable to the journal for publication. Other expressions in methods and results sections are very similar to the authors’ previous work that should be changed and re-paraphrased. The reviewer thinks the concept and approach are appropriate for further research but it needs to be considered again for publication.
Author Response
Dear reviewer,
We really appreciate your useful comments and suggestions on our manuscript. According to the comments and suggestions, we have checked and modified the manuscript carefully. The detailed corrections are listed below. We have marked the changes in the revised manuscript in red color. We highly appreciate your attention and consideration on our work.
Sincerely,
Dr. Huayong Zhang
Point 1: This manuscript presents that the effects of Cd combining with Fe and Ni on anaerobic fermentation with corn stover combined with cow manure. This would be an interesting manuscript for readers of International Journal of Environmental Research and Public Health Journal, but this study is very similar to the authors’ previous journal published August 2019, titled “Effect of Cd and Zn mixture on the biodegradation and microbial communities of anaerobic fermentation process.” The reviewer believes that the concept, methods, technical approach, many expressions of the sentences and paragraphs in the previous manuscript and current works are overlapped that dilute the novelty of the work. For example, the results of table 1 is almost same with few changes in the words. This is not acceptable to the journal for publication. Other expressions in methods and results sections are very similar to the authors’ previous work that should be changed and re-paraphrased. The reviewer thinks the concept and approach are appropriate for further research but it needs to be considered again for publication.
Response 1: Thank you very much for your comments and suggestions. We modified the manuscript according to the comments. Comparing with the previous paper we published “Effect of Cd and Zn mixture on the biodegradation and microbial communities of anaerobic fermentation process”, we would like to highlight the novelty of this study by the following aspects:
(a) Since the same raw materials were used for experiments, we cited the previous article in the M&M section and removed Table 1 from the modified manuscript to avoid repetition.
(b) Fe and Ni are also the commonest distributed metals in biomasses. They have different impacts and combining effect with Cd on fermentation process comparing with Zn. The studies on anaerobic fermentation by the addition of Cd + Fe and Cd + Ni in the literatures are lack. The research objectives and the results of the present study are different from the previous publications. In the revised manuscript, more detailed discussion about the different impacts of Fe and Ni combining with Cd on biogas production, pH variation and VFA utilization were added in the Results and discussion section.
(c) In the above-mentioned publication, the impacts of the metals were focused on the lignocelluloses degradation and the variation of organic matters during the process (as indicated by COD), i.e. the hydrolysis stage was investigated and analyzed. However, the present study is more interesting in the generation and consumption of VFA as well as its relationships with the microorganisms during the anaerobic fermentation process.
Therefore, we believe that the novelty of the present study is not overlapped after the modification.

Round 2
Reviewer 1 Report
Please correct minor spacing and punctuations. Some sentences do not have period.
Author Response
Dear reviewer,
We really appreciate your useful comments and suggestions on our manuscript. According to the comments and suggestions, we have checked and modified the manuscript carefully. The detailed corrections are listed below. We have marked the changes in the revised manuscript using "Track Changes" function in Microsoft Word. We highly appreciate your attention and consideration on our work.
Sincerely,
Dr. Huayong Zhang
Point 1: Please correct minor spacing and punctuations. Some sentences do not have period.
Response 1: Thank you very much for your suggestions. We checked and corrected manuscript carefully, including space, punctuations, period etc.

Reviewer 3 Report
The revised version has been improved and presented the differences between the previous work and the current work with summarized data. This will be very interesting for the readers of International Journal of Environmental Research and Public Health. The reviewer has a couple of comments that:
1) The authors need to replace the word "dung" to "waste" or "manure"
2) There are minor errors in spells, font sizes and formatting in the manuscript that should be revised before submitting again.
Author Response
Dear reviewer,
We really appreciate your useful comments and suggestions on our manuscript. According to the comments and suggestions, we have checked and modified the manuscript carefully. The detailed corrections are listed below. We have marked the changes in the revised manuscript using "Track Changes" function in Microsoft Word. We highly appreciate your attention and consideration on our work.
Sincerely,
Dr. Huayong Zhang
Point 1: The authors need to replace the word "dung" to "waste" or "manure"
Response 1: Thank you very much for your suggestions. We replaced the word “dung” with “manure” in the revised manuscript.
Point 2: There are minor errors in spells, font sizes and formatting in the manuscript that should be revised before submitting again.
Response 2: Thank you very much for your suggestions. We checked and modified the spelling, font sizes and format in the revised manuscript.
